# Neuropsychiatric and Neuropsychological Aspects of Alcohol-Related Cognitive Disorders: An In-Depth Review of Wernicke’s Encephalopathy and Korsakoff’s Syndrome

**DOI:** 10.3390/jcm12186101

**Published:** 2023-09-21

**Authors:** Lucian Eva, Felix-Mircea Brehar, Ioan-Alexandru Florian, Razvan-Adrian Covache-Busuioc, Horia Petre Costin, David-Ioan Dumitrascu, Bogdan-Gabriel Bratu, Luca-Andrei Glavan, Alexandru Vlad Ciurea

**Affiliations:** 1Faculty of Medicine, Dunarea de Jos University, 800010 Galati, Romania; elucian73@yahoo.com; 2Department of Neurosurgery, Clinical Emergency Hospital “Prof. Dr. Nicolae Oblu”, 700309 Iasi, Romania; 3Department of Neurosurgery, Clinical Emergency Hospital “Bagdasar-Arseni”, 041915 Bucharest, Romania; 4Department of Neurosurgery, “Carol Davila” University of Medicine and Pharmacy, 020021 Bucharest, Romania; razvan-adrian.covache-busuioc0720@stud.umfcd.ro (R.-A.C.-B.); horia-petre.costin0720@stud.umfcd.ro (H.P.C.); david-ioan.dumitrascu0720@stud.umfcd.ro (D.-I.D.); bogdan.bratu@stud.umfcd.ro (B.-G.B.); luca-andrei.glavan0720@stud.umfcd.ro (L.-A.G.); prof.avciurea@gmail.com (A.V.C.); 5Department of Neurosciences, “Iuliu Hatieganu” University of Medicine and Pharmacy, 400012 Cluj-Napoca, Romania; 6Neurosurgery Department, Sanador Clinical Hospital, 010991 Bucharest, Romania

**Keywords:** Wernicke’s Encephalopathy, Korsakoff’s Syndrome, alcohol-related cognitive disorders, neuropsychiatric symptoms, neuroimaging research

## Abstract

Alcohol-related cognitive disorders have long been an area of study, yet they continue to pose challenges in the diagnosis, treatment, and understanding of underlying neuropsychiatric mechanisms. The present article offers a comprehensive review of Wernicke’s Encephalopathy and Korsakoff’s Syndrome, two conditions often seen on a continuum of alcohol-related brain damage. Drawing on current medical literature, neuroimaging studies, and clinical case reports, we explore the neuropsychiatric and neuropsychological profiles, symptomatology, and differential diagnoses of these disorders. We delve into the biochemical pathways implicated in the development of WE and KS, notably thiamine deficiency and its impact on neurotransmitter systems and neural networks. The article also addresses the challenges in early diagnosis, often complicated by non-specific symptoms and co-occurring psychiatric conditions. Furthermore, we review the current state of treatment protocols, including pharmacological and non-pharmacological interventions. Finally, the article highlights gaps in current knowledge and suggests directions for future research to improve diagnosis, treatment, and patient outcomes. Understanding the nuanced interplay between the neuropsychiatric and neuropsychological aspects of WE and KS is crucial for both clinicians and researchers alike, in order to provide effective treatment and to advance our understanding of these complex conditions.

## 1. Introduction

### 1.1. Wernicke’s Encephalopathy (WE) and Korsakoff’s Syndrome (KS): Brief History and Linkage

Carl Wernicke first described WE, now named in his honor. This condition featured symptoms like lethargy, ophthalmoplegia, ataxia and cognitive impairment. At roughly the same time, but independently from Wernicke’s, work Sergei Korsakoff presented his doctoral thesis entitled “Alcoholic Paralysis”, discussing a particular form of memory loss found among chronic alcoholism cases known as circumscribed amnesia.

Over fifty years later, it took an unusually long time for researchers to uncover a shared cause between these two disorders: thiamine deficiency was identified as being at the core of both conditions, contributing to their symptoms in both WE and KS and becoming recognized as their link.

Now we understand more fully the temporal relationship between WE and KS, with two distinct phases:WE typically presents with acute/phase lesions to areas such as the periventricular nuclei, thalami and components of Papez’s circuit (including mammillary bodies).Korsakoff’s Syndrome progresses into its chronic phase with lesions developing into more permanent bilateral damage that causes global amnesia.

Noting this correlation, the combined condition is often referred to as Wernicke-Korsakoff Syndrome (WKS). Amnesia played an instrumental role in connecting WKS to neuropsychology; further exploration into memory processes as well as identification of distinct neural substrates responsible for different aspects of memory function ensued from this linkage [1].

KS often develops in those who have experienced WE but failed to receive appropriate thiamine replacement therapy promptly and appropriately. The most recognizable feature of KS is an amnesia which may be profound; combined with additional cognitive and behavioral impairments that often appear with more serious cases, its effects can significantly impact an individual’s daily functioning and quality of life [2]. Research endeavors examining the scope, arrangement and essence of episodic memory impairments within KS have significantly advanced our understanding of memory as an abstract concept. Furthermore, these studies have shed light on memory being not an all-inclusive function, and have demonstrated its multidimensional nature. Furthermore, examination of KS has highlighted diencephalic structures’ vital importance to memory processes—leading researchers into further exploring distinct brain structures or neural pathways responsible for individual memory processes to create more intricate mnemonic components [3].

### 1.2. Connection to Alcohol-Related Cognitive Disorders

KS and WE are two distinct neurological conditions; however, they frequently share similarities due to a deficiency of thiamine (vitamin B1) which often stems from chronic alcohol misuse. Current diagnostic frameworks for cognitive impairment related to alcohol consumption can be divided into two primary syndromes: Wernicke-Korsakoff syndrome (WKS) and alcohol/related dementia (ARD) [4].

WE can be identified clinically by three distinct symptoms, including changes to mental status, dysfunction in oculomotor capabilities and issues with cerebellar function. Note, however, that not all patients exhibit all three symptoms at once. According to Western studies, up to 90% of WE cases can be linked to alcohol misuse [5]. Left undiagnosed or untreated, WE can progress into KS, which occurs in 56–84% of those with an alcoholism history and possibly less frequently among cases unrelated to alcohol consumption. The main cognitive manifestation of KS is intense amnesia, characterized by difficulties forming new memories (anterograde) and recalling old ones (retrograde), although it can also produce other cognitive and behavioral symptoms [6]. Alcohol-related dementia (ARD), as an identifiable clinical entity, has generated much discussion, due to the uncertainty regarding its causes, lack of diagnostic criteria and difficulties assessing affected populations. Due to this lack of clarity regarding ARD classification as a distinct clinical entity, some scholars prefer the term ‘alcohol-related brain damage’ which encompasses neurocognitive impairments caused by chronic alcohol consumption such as Wernicke-Korsakoff Syndrome and ARD [4].

Uncertainty remains as to whether alcohol-related dementia (ARD) results directly from neurotoxic effects of alcohol consumption (neurotoxicity hypothesis), alternative pathologies like thiamine deficiency or from multiple interlinked factors. Oslin et al. have developed preliminary diagnostic criteria for ARD. Their five-year period with men drinking an average of 35 standard drinks weekly (30 for women) over this five-year period should be enough to induce this disorder [7].

Diagnosing alcohol-related dementia (ARD) can be complex, due to other related conditions, including head trauma, substance abuse, co-occurring psychiatric disorders, and elevated cardiovascular risk factors. Although neuropsychological characteristics of ARD have received limited study, observations indicate a more comprehensive cognitive decline compared to WKS [4].

This comprehensive review covers many facets of alcohol-related cognitive disorders. These aspects include diagnostic criteria, the cognitive aftermath and the neuropathology underlying these disorders, and concurrent physical health conditions and trends in epidemiology, as well as available treatment approaches, the impact on quality of life for affected individuals, mental capacity considerations, and the role neuroimaging plays in both diagnosis and management.

This comprehensive article’s overall aim is to serve as an indispensable reference for researchers, healthcare practitioners, and clinicians. Its objective is to illuminate various dimensions of WE and Korsakoff’s Syndrome so as to advance early-detection practices, enhance patient outcomes, and raise awareness regarding these debilitating conditions.

## 2. Assessment and Diagnostic Work-Up

Criteria for Diagnosing WE: Diagnosing WE can be challenging, given the various ways it manifests clinically; however, the latest diagnostic criteria suggest at least two of four signs should be present—diet inadequacy, abnormalities in oculomotor function abnormalities, deficits in cerebellar function and altered mental state or mild impairment of memory are indicators. While not all patients will necessarily exhibit all these classic signs, an elevated sense of suspicion is necessary in order to accurately confirm diagnosis [8].

Caine et al. [8] recently developed operational criteria for the clinical diagnosis of WE called the Caine criteria (Table 1). While these criteria could greatly facilitate diagnosis, histopathological evidence still forms the cornerstone of WE diagnosis [9].

2.Criteria for Diagnosing Korsakoff’s Syndrome (KS): Recognizing Korsakoff’s Syndrome, often the result of chronic thiamine deficiency or WE, requires the identification of certain markers. These include both anterograde (inability to form new memories) and retrograde amnesia, with difficulty recalling past memories; anterograde amnesia is often accompanied by confabulations (fabricated or altered memories used to fill memory gaps), which cannot be attributed to any other medical cause; furthermore patients might exhibit executive dysfunction or have limited awareness regarding memory impairments affecting them.

At present, there is a lack of consensus on a universal definition or diagnostic criteria for Korsakoff’s Syndrome (KS) [10]. According to the *Diagnostic and Statistical Manual of Mental Disorders, Fifth Edition (DSM-5)*, KS is characterized as “an alcohol-induced major neurocognitive disorder with amnestic confabulatory features” [11]. This current classification is problematic, since it erroneously presumes that the etiology of KS is exclusively alcohol-related, thereby limiting its diagnostic scope since it fails to establish a direct relationship with Wernicke’s Encephalopathy (WE). *The International Classification of Diseases, Tenth Edition (ICD-10)*, offers a separate categorization for alcoholic and non-alcoholic forms of KS, but this further complicates the diagnostic landscape. Non-alcoholic forms are categorized under organic mental disorders, whereas those induced by alcohol abuse fall under the F10.26 code category, thereby rendering the application of ICD-10 criteria more cumbersome [12].

One objective of the present research was to formulate an exhaustive definition and outline potential diagnostic criteria for KS [2].

### 2.1. Neuroimaging

Early neuroradiological studies of WE from the 1970s relied on computed tomography (CT) to identify third ventricle enlargement, but struggled with limitations in detecting edema or localized damage [13,14]. The advent of magnetic resonance imaging (MRI) revolutionized this field, offering higher sensitivity to tissue-water content [15]. MRI was more effective in diagnosing WE, identifying abnormalities in 7 of 15 studied WE patients compared to just 2 of 15 through CT. It also highlighted further affected regions like the periaqueductal gray matter and mammillary bodies [16]. MRI’s sensitivity thus underscores its importance in detecting subtle neuropathological changes, especially in alcohol abusers without overt symptoms of WE.

Initial MR techniques for enhancing the visibility of edematous lesions in WE focused on T2-weighted late-echo sequences. Bilateral hyperintensity in areas like mammillary bodies and thalamic nuclei is a key neuroradiological feature of acute WE, observed in both alcoholic and nonalcoholic cases [17,18]. These MRI findings are consistent with postmortem reports [19,20]. Neuropathology affecting both brain hemispheres could exacerbate clinical symptoms. Although less common in WE, a specific MR study noted excess fluid in the central pons of patients with Wernicke-Korsakoff Syndrome (WKS), correlating with markers like macrocytic anemia and cognitive fluency in non-WKS alcoholic patients [21].

The FLAIR (MR fluid-attenuated inversion recovery) sequence marked a significant technological leap from traditional T2 methods by incorporating additional T1 contrast mechanisms. FLAIR effectively suppresses signals from cerebrospinal fluid (CSF), enhancing the visibility of edematous tissue regions [22]. One case study involving hyperemesis gravidarum documented high signal intensity in the mammillary bodies and hypothalamus, which subsided following thiamine treatment [23]. FLAIR imaging of nonalcoholic WE cases showed hyperintense signals in specific brain regions. Follow-up assessments indicated recovery in cases without prior cortical damage, while no improvement was observed in those with existing damage [17].

Magnetic resonance diffusion-weighted imaging (DWI) has shown high accuracy in detecting brain pathology related to WE. Interestingly, what would intuitively appear as high diffusivity levels actually show up as hyperintense on DWI due to a phenomenon known as the T2 shine-through effect [24]. This counterintuitive brightness represents low, rather than high, diffusivity levels.

Unlu et al. (2006) found that WE often features abnormalities in periventricular and thalamic tissues [25]. A case study showed that a distinct DWI bright signal originated from abnormally low diffusivity in the cerebellum, which improved with thiamine treatment but left lingering motor impairment [26]. Acute WE studies consistently reveal increased DWI signal intensity and decreased diffusivity as captured via apparent diffusion coefficient (ADC) images [26]. Although DWI signals can be affected by the T2 shine-through effect, combining DWI with ADC imaging allows for a more comprehensive characterization of WE lesions as they evolve from high- to low-diffusivity states.

### 2.2. Biomarkers

Alcohol-related health complications are widespread, yet frequently go undetected. Precise screening for alcohol misuse hinges on tests that can both sensitively and specifically spot those engaging in dangerous drinking habits and track their behavior. By coupling laboratory indicators like AST, ALT, MCV, and GGT with self-reported data, it becomes feasible to identify and monitor those engaging in harmful drinking patterns [27].

While much research on alcohol dependence has primarily centered on men, studies tailored to the unique experiences of women underscore their heightened susceptibility. This vulnerability stems from differences in physiology, metabolism, hormonal interactions, and additional concerns during pregnancy [28].

It is worth spotlighting several key alcohol biomarkers:Blood alcohol concentration (BAC): Determined from samples of blood, breath, saliva, or urine, BAC can confirm alcohol intoxication. Notably, men and women often display significant variances in their BAC levels.Gamma-glutamyltransferase (GGT): A spike in GGT signals potential early-stage liver damage from alcohol. It is a useful marker for alcohol-triggered harm, although it might not be as effective in discerning dangerous drinking habits in younger individuals and women.Transaminases AST and ALT: A rise in ALT points to liver damage. The AST/ALT ratio serves as a tool for identifying alcohol-related liver diseases, though its accuracy can waver across different age segments.Mean corpuscular volume (MCV): a heightened MCV can be an indicator of consistent excessive drinking, particularly in women.Carbohydrate-deficient transferrin (CDT): this marker is adept at indicating heavy alcohol consumption but may not be as effective for women, especially those expecting.Fatty acid ethyl esters (FAEEs): they efficiently differentiate between casual and excessive drinkers and can pinpoint alcohol consumption during pregnancy.Ethyl glucuronide (EtG): this marker excels at detecting excessive drinking instances, even if other traditional indicators might miss them.Phosphatidylethanol (PEth): a highly specific and sensitive sign of prolonged alcohol consumption, PEth can be detected in the blood for up to two weeks post drinking.Thiamine and its esters: Chronic alcohol users often have a thiamine deficiency. By directly measuring thiamine and its esters in red blood cells, this deficiency can be gauged.

Leveraging these biomarkers can offer invaluable data on alcohol misuse, subsequently facilitating diagnosis and treatment, especially for women and during their pregnancies.

Even though many of these markers are used to determine the health of the liver in general, they should be taken into consideration when trying to diagnose WE or KS. Markers like CDT, EtG, FAEEs and Peth should also be kept in mind when trying to diagnose these two diseases, being of great value in interpreting the etiology of the liver damage, and thus detecting alcoholism. 

To summarize, pinpointing conditions like WE and KS involves a comprehensive evaluation of clinical symptoms, cognitive tests, and neuroimaging to detect the brain anomalies tied to thiamine deficiency. The relentless pursuit of dependable biomarkers and innovative tools is poised to significantly boost early detection and management strategies for these intricate neurological conditions [29]. 

## 3. Neuropsychiatric and Cognitive Sequelae

As previously highlighted, WE and KS are typically perceived as being interrelated, since both originate from a deficiency in thiamine. Consequently, medical professionals should simultaneously assess both conditions. While WE is conventionally associated with a combination of cognitive disturbances, ophthalmoplegia, and ataxia, it is worth noting that less than a fifth of patients will manifest all these symptoms [30].

WE patients frequently exhibit impairments in neuropsychological functioning that become increasingly evident as their physical condition improves. Neuropsychological functioning encompasses various cognitive abilities ranging from fundamental processes like attention and concentration to more intricate ones like memory, executive functioning, and reasoning; all these higher-order cognitive processes play a key role in overall quality of life for an individual, although unfortunately clinicians sometimes overlook this assessment [31]. 

When delving deeper into KS, in contrast to Wernicke’s encephalopathy, specific cognitive deficiencies surface. These encompass anterograde amnesia, which severely hampers the ability to assimilate new information, and retrograde amnesia, as well as executive function disorders leading to reduced self-restraint and challenges in areas like judgment, strategizing, and problem resolution [32]. A key manifestation of anterograde amnesia is confabulation, where individuals unconsciously fill memory gaps with fabricated details [33].

Assessing memory function—an essential factor for maintaining quality of life (QOL)—was conducted via PGI-BBD evaluation once the physical condition had improved. Korsakoff memory impairments were evident immediately upon assessment, correlating with initial assessments; however, cases where assessments could take place post improvement proved challenging, due to symptoms overlapping globally with memory dysfunction and memory deficits; pinpointing exactly when cognitive confusion recedes, paving the way for memory deficits, was often difficult and required in-depth discussion amongst participants.

Memory dysfunction was so debilitating in these initial cases that these individuals experienced difficulty maintaining an acceptable quality of life (QOL), accompanied by noticeable occupational impairments. Early administration of thiamine led to improvements in ataxia and ophthalmoplegia; however, confusion and neurological dysfunction persisted for an extended period, according to neuropsychological assessments [34], even when confusion eventually resolved itself in later phases; deficits in memory function, as well as the capacity for learning new material, persisted for an extended timeframe.

In KS, while anterograde amnesia stands out as a major symptom, the patient’s remote memory is also impacted. This leads to retrograde amnesia that affects both general knowledge—such as facts from the news—and personal autobiographical events [35]. For those with KS, their memory loss can span numerous years or even multiple decades, though memories from early life, like childhood, tend to remain intact [36]. This pattern of losing more recent memories while retaining distant ones in KS and other conditions has been termed a “temporal gradient”. Théodule Ribot first documented this observation in 1881, leading to its designation as Ribot’s law. While many studies on KS identify a pronounced temporal gradient, some suggest an even memory impairment across all past timelines [37]. These differing findings may arise from methodological challenges or the type of memory being examined: general vs. personal [38]. Autobiographical episodic memories entail personal experiences set in specific places and times, like recalling the exact moment of meeting a significant other on a balmy evening at a poolside bar. Fewer studies focus on these autobiographical memories, due to their complex nature compared to general memories. A recent investigation by Rensen et al. [38] confirmed that the temporal gradient also influences episodic autobiographical memory, supporting earlier research findings.

WE and Korsakoff’s Syndrome can have severely disabling consequences. Impaired memory and executive function make even basic daily tasks daunting for patients. This impairment often results in challenges in sustaining relationships and jobs. The hindered ability to recall new events or acquire new skills drastically affects their independence and life quality. Emotional health deteriorates as patients confront their cognitive restrictions and the loneliness stemming from social disconnect [2].

In essence, both WE and Korsakoff’s Syndrome arise primarily from a thiamine deficiency, frequently linked to prolonged alcohol consumption. These conditions present with psychiatric symptoms and profound cognitive deficits, and significantly hinder daily activities and overall life satisfaction. Prompt identification, thiamine replenishment, and holistic rehabilitation are vital to counteract these conditions’ effects.

## 4. Neuropathology of Wernicke’s Encephalopathy and Korsakoff’s Syndrome

Thiamine is prevalent in various parts of the body, notably in skeletal muscles, liver, heart, kidney, and brain. The balance of thiamine in the body is maintained through adequate dietary intake, absorption in the intestines, reuptake in the kidneys, and storage and release in the liver, when necessary. The brain has a safety buffer of thiamine, and noticeable neuropsychiatric symptoms only emerge when thiamine levels drop below 20% of the usual amount [39].

The primary active form of thiamine in the central nervous system is thiamine pyrophosphate (TPP). It acts as a critical component or cofactor for three significant enzymes in glucose metabolism: transketolase, pyruvate dehydrogenase, and α-ketoglutarate dehydrogenase. These enzymes participate in crucial metabolic processes like the pentose phosphate pathway, glycolysis, and the citric acid cycle. These processes produce vital molecules for functions in neurons and glial cells, such as nucleic acids, neurotransmitters, myelin, and energy-rich compounds like ATP [40]. Moreover, because of its involvement in cellular energy processes, thiamine aids in defending cells against oxidative stress. In its triphosphorylated form (TTP), thiamine is essential for nerve membrane functionality (Figure 1).

### Thiamine Deficiency in Relation to Alcohol

Chronic alcohol consumption can lead to thiamine deficiency due to several reasons:

Alcoholics often opt for diets high in carbohydrates but low in vitamins [41].Without proper supplementation, thiamine reserves are exhausted within 2–3 weeks.Acute alcohol consumption can hamper thiamine absorption from the gut.Alcohol’s impact on renal epithelial cells results in greater thiamine loss through the kidneys.Chronic alcoholic liver disease can slash the liver’s thiamine storage capability by up to 73%.Alcohol can reduce the enzymatic activity of thiamine pyrophosphokinase (TPK), which in turn lowers the amount of available TPP.Reduced TPK activity further impedes the facilitated diffusion of thiamine into cells.Alcohol can decrease the absorption of thiamine produced by colonic bacteria.Many alcoholics have hypomagnesemia, a deficiency of magnesium, which is a vital cofactor in thiamine utilization.

In essence, alcohol can induce a state of thiamine deficiency through both direct and indirect mechanisms [42].

In situations where Wernicke–Korsakoff Syndrome (WKS) arises without alcohol as a factor, thiamine deficiency can be attributed to one of these four reasons: limited supply, hindered usage, increased consumption, or augmented loss of thiamine. Limited supply often happens during times of starvation, malnourishment, poor absorption, excessive losses, or vomiting. Hindered usage takes place when the body struggles to process thiamine due to diminished enzyme activity or coenzyme deficiencies. Thiamine consumption increases when there is a spike in the body’s glucose metabolism, such as in instances of heightened metabolic rate, enhanced carbohydrate metabolism, or situations with swift cell regeneration. Interestingly, even when on a diet considered sufficient, thiamine deficiency can still occur under these conditions. Specifically, in the context of cancers, particularly the types associated with rapid cellular growth like leukemia and lymphoma, there is a surge in thiamine consumption [43]. Cancer can often lead to a reduced thiamine intake, either due to the disease’s direct effects, loss of appetite, chemotherapy side effects, or vomiting (see Table 2) [44]. If patients are started on total parenteral nutrition (TPN) without the necessary thiamine supplementation, they can end up being deficient. Lastly, augmented thiamine loss is observed in hemodialysis, since thiamine is expelled into the dialysate [45].

For an in-depth exploration of the physiological impacts of thiamine deficiency, refer to the work by Sechi and Serra. When the brain experiences a lack of thiamine, it can result in cell-damaging swelling and an increase in astrocyte volume within just 4 days. Between the 7th and 10th day, a drop in transketolase activity brings about dysfunction in endothelial cells, leading to the production of nitrous oxide and the spilling of intracellular glutamate into surrounding spaces. This results in disrupted osmotic balances and the generation of free radicals, further causing swelling and increased permeability in the blood-brain Barrier (BBB). By the time 14 days of thiamine deficiency have passed, the damage to neuronal DNA and increased lactic acid lead to permanent structural harm and neuronal cell death [46]. During the process of alcohol withdrawal, the heightened sensitivity of the NMDA-receptor can compound the neurotoxic effects, releasing even more glutamate. Some brain regions are more susceptible to these damaging effects than others. For instance, in one study, the mammillary bodies displayed signs of damage in every examined case. However, the reasons for the particular vulnerability of areas like the mammillary bodies, the region around the aqueduct, and the tectum are not entirely clear, as factors like the embryonic origin of cells, blood flow patterns, and tissue characteristics do not seem to provide a full explanation. As scientists delve deeper into the molecular mechanisms behind cell damage resulting from thiamine deficiency, the reasons behind the selective harm to specific brain regions, including the mammillary bodies, remain a mystery [47].

## 5. Somatic Comorbidity and Epidemiology

If WE is suspected, immediate treatment is crucial. On average, symptoms of this condition, such as lethargy, confusion, and difficulty walking that may lead to a fall, manifest around 3 to 4 days prior to a subsequent hospitalization [48]. However, within a hospital setting, WE can be triggered by refeeding syndrome in just 2 to 3 days [49]. This is because most patients with thiamine deficiency may have neglected proper nutrition for months, and in some cases, might not have consumed any food for days or even weeks.

For such patients, it is essential to reintroduce calories slowly and under the guidance of a dietitian. Continuous monitoring is necessary in the initial days of hospitalization, and includes regular checks of blood glucose levels and electrolytes. Refeeding syndrome is typified by imbalances in water and electrolytes, especially low levels of phosphorus, potassium, and magnesium, alongside glucose intolerance, signs of thiamine deficiency, and excessive fluid retention [50]. Administering glucose or reintroducing carbohydrates without supplementing thiamine can pose a risk of triggering Wernicke encephalopathy, particularly in individuals with already low thiamine levels.

Criteria that indicate a high likelihood of developing refeeding syndrome can be found in the recommendations set forth by the American Society for Parenteral and Enteral Nutrition (ASPEN). If no abnormal laboratory results are observed for elements like phosphate, calcium, potassium, magnesium, and glucose, monitoring in relation to refeeding syndrome can cease after 72 h.

Various other risk factors can also play a significant role in leading to thiamine deficiency. These include infections, esophageal narrowing (Barrett’s esophagus), colitis, and, notably, the renal loss of thiamine experienced in conditions like diabetes mellitus or nephropathy [51]. Loss of appetite and vomiting can both result from and exacerbate thiamine deficiency [52].

### Examination of Epidemiological Data and Demographics

There is a limited amount of epidemiological information regarding cognitive disorders tied to alcohol. To truly grasp the impact of diseases like Wernicke-Korsakoff syndrome (WKS) and alcohol-related dementia (ARD), it is essential to have updated data on their incidence and mortality rates within populations [53].

The current knowledge about the frequency and occurrence of WE and alcohol-related dementia (ARD) has significant gaps. Most of our insights into these conditions are based on research from the 1970s and 1990s, which might now be considered outdated. Autopsy-based data from the past indicated that WE impacted between 0.4% and 2.8% of the overall population, and an alarming 12.5% to 59% of those with alcoholism or deaths related to alcohol consumption [54]. However, modern research is not without its own challenges in methodology and reach. For KS, data from the Netherlands estimated a prevalence of 3–4.8 per 10,000 people, based on healthcare records. Meanwhile, data from Glasgow suggested an annual KS rate of 12.5 to 81.25 per million inhabitants between 1990 and 1995.

The frequency of ARD shows significant variance across studies. Some assessments derive their estimates from associating dementia rates with alcohol use patterns or by examining particular groups. Among various research studies, ARD prevalence varied wildly, from 0.119% in general hospital stays to a staggering 25.6% in elderly patients with alcoholism being treated in substance abuse clinics [55]. Broader population-focused research indicated prevalence rates as low as 0.0066% in those aged 30–64 and as high as 0.7% in US Medicare beneficiaries aged 68 or older.

The future outcomes and prolonged mortality trends of both WKS and ARD remain underexplored. A previous study examining 245 WKS patients reported an immediate mortality rate of 17%, with subsequent deaths primarily attributed to infections, liver diseases, and cancers [56]. Contemporary studies highlight an acute mortality rate of 5.3% to 10% for those diagnosed with WE. Hospital patients diagnosed with alcohol-linked WE or KS exhibited a median survival rate of 8 years and a death rate of 7.4 per 100 person-years. The predominant causes of death were bacterial infections and cancers [55].

## 6. Pharmacological and Nonpharmacological Treatments

Alcohol impacts the gut in a way that unpredictably hinders thiamine absorption. To ensure an adequate concentration of thiamine in the bloodstream, it should be given through injection or infusion. The reasoning for elevated blood levels of thiamine links back to how it is transported to the central nervous system (CNS). As noted, thiamine’s secondary method of reaching the CNS, through simple diffusion, does not meet the expected quantity. Theoretically, with increased serum levels, thiamine can penetrate the CNS more effectively [57].

The exact definition of a “high dose” of thiamine remains undefined. Back in 1950, Victor and Adams opted for 100 mg of thiamine to combat WE, using their best judgment of what might be considered a high dose [58]. Some experts advise treating alcoholic WE patients with 500 mg or more, although this suggestion is built on sparse evidence. The sole published randomized trial evaluating thiamine for treating WKS contrasted five thiamine doses (ranging from 5 to 200 mg daily) over two days for patients undergoing alcohol withdrawal treatment [59]. Those on the highest dose showcased superior performance in delayed alternation, a task pinpointing WKS-related cognitive issues. Yet the findings did not exhibit a consistent dose–response relationship. An updated 2008 Cochrane review concluded that evidence is too scant to give clear guidance on thiamine’s dosage, frequency, delivery method, or duration for treating or preventing WKS in alcoholics.

Both the Royal College of Physicians (RCP) and the European Federation of Neurological Societies have released guidelines on preventing WE in alcoholics (refer to Table 3). Despite the distinctions between them, both emphasize the need for vigilance when it comes to WE and advocate for robust treatment. This stance arises from the diagnostic challenges in a clinical context and the favorable risk–benefit profile of using high thiamine doses [60].

The safety profile of intravenous (IV) thiamine was examined in a study where 1070 doses of 100 mg thiamine hydrochloride were rapidly given to 989 patients [61]. Minor reactions, including transient arm burning lasting from a few seconds to minutes occurred in about 1.02% of cases. One major reaction was observed, which was widespread itching without other symptoms, occurring at a rate of 0.093%. Although there have been documented cases of severe allergic reactions to IV thiamine, the likelihood is extremely low, with a recorded risk of one reaction per five million thiamine vials in the UK [42]. A survey from emergency department doctors estimated that out of around 300,000 patients receiving thiamine through injection, none experienced severe allergic reactions. In contrast, there is a 1–10% risk of allergic reactions to penicillin, a 2–3% risk of a reaction to contrast agents, and a 1–18% risk of reaction to streptokinase. Hence, thiamine is typically regarded as safe for injectable use [62].

For alcoholics, replenishing magnesium is crucial, as their diminished magnesium levels can render thiamine treatment ineffective.

The appropriate timing for administering thiamine in relation to carbohydrate has been a topic of discussion. Given thiamine’s role in glucose metabolism, a spike in metabolic rate can lead to a relative thiamine deficiency, potentially triggering WKS. Several case studies and animal experiments have reported such outcomes [63]. However, a recent review of WKS emergency management posits that a single dose of glucose will not induce this condition, suggesting that in emergencies glucose administration should not be postponed because of thiamine [58]. 

### 6.1. Effectiveness and Limitations of Nonpharmacological Interventions

Emerging evidence is challenging the long-standing view that Korsakoff’s Syndrome (KS) is an unmodifiable condition resistant to further cognitive recovery. This paradigm shift is bolstered by a growing body of research advocating the efficacy of memory rehabilitation in KS patients [32]. Among various interventions, compensatory memory-enhancement techniques, which encompass traditional aids like agendas and memory cards, as well as digital tools such as smartphones and smartwatches, appear to be particularly promising. A meta-analysis of six studies highlights the fact that these techniques are most effective when (1) the therapeutic objectives are clearly defined, (2) adequate time is committed to individualized patient instruction, and (3) these aids are cohesively integrated into comprehensive learning methodologies.

In this context, errorless learning emerges as a theoretically ideal approach that aligns well with the unique cognitive strengths and vulnerabilities of KS patients. The core principle of this strategy lies in minimizing the opportunity for errors during the acquisition phase of learning. By eliminating the margin for guesswork, errorless learning fortifies the patient’s procedural memory system against the incorporation of maladaptive or erroneous strategies. This is particularly crucial for KS patients whose episodic memory impairments would otherwise be incapable of correcting such errors [64]. Although empirical studies directly comparing errorless learning with traditional trial-and-error methods are sparse and yield mixed results, the technique holds promise, especially within specific learning contexts. Importantly, the utility of errorless learning extends beyond the domain of procedural memory, potentially offering advantages in the realm of semantic memory as well. Given that KS patients often suffer from episodic memory deficits, the errorless learning approach compensates by reinforcing and leveraging more robust procedural and semantic memory systems, thereby providing a more holistic strategy for cognitive rehabilitation.

### 6.2. Emerging Therapies and Future Directions

#### 6.2.1. Thiamine in the Treatment and Prevention of Wernicke-Korsakoff Syndrome for Alcohol Abusers

While WKS is conventionally treated with thiamine once diagnosed, the efficacy of this treatment, especially regarding cognitive symptoms, remains uncertain. Current guidelines concerning the dosage and treatment duration with thiamine are largely based on best guesses. We aimed to find randomized controlled trials that either contrasted thiamine with placebos or other treatments, or assessed varying thiamine treatments. From our search, two studies met the prerequisites for inclusion. However, one did not provide any analyzable data, and the other’s analysis was hampered by design flaws and inadequate data presentation. Consequently, there is an absence of solid evidence from these trials to guide doctors on the optimal thiamine treatment parameters to prevent or address WKS stemming from alcohol misuse [65].

#### 6.2.2. How the Intervention Might Work

This analysis delves into WKS resulting from alcohol abuse, which is the predominant cause in developed nations. Between 30% and 80% of those abusing alcohol exhibit either clinical or biochemical indications of a thiamine deficit. Notably, alcohol seems to amplify the thiamine quantity essential for effective treatment compared to cases where the deficiency stems primarily from nutritional reasons [66]. Even with data pointing towards a significant undetected neuropathological impact due to thiamine shortage, comprehensive studies evaluating thiamine’s therapeutic potency in alcohol abusers remain scant. Rapid improvements in ataxia and eye movement anomalies with thiamine administration have been noted, yet its influence on memory remains ambiguous. One comprehensive look into this area revealed the absence of systematic, placebo-controlled studies exploring the usage of injectable B-complex vitamins (thiamine-inclusive) in alcohol abuse scenarios. Hence, before suggesting varied treatment strategies for those at risk of WE and those currently afflicted by it, more research is imperative [67]. 

## 7. Quality of Life and Mental Capacity

The term quality of life (QoL) is frequently associated with an individual’s overall well-being. Despite the apparent increase in patients with KS, there is a significant gap in our understanding of their QoL. Presently, most insights are derived from dementia care. This study aims to juxtapose various QoL facets in KS patients against those with dementia from identical care settings, in a bid to deepen our grasp on the socio-emotional dimensions of KS [68]. 

Measurements: The QUALIDEM scale was employed to evaluate Quality of Life (QoL). To contrast the QoL of patients with KS against those with dementia, multivariate linear regression analysis was utilized, considering the variables “age”, “gender”, and “nursing home”.

Results: Out of the 147 participants, 72 (48.9%) had a KS diagnosis. KS patients exhibited a higher overall QoL. On the QUALIDEM subscales, KS patients outperformed dementia patients in areas such as “Restless tense behavior”, “Social relations”, and “Having something to do”. There was a noticeable inclination for KS patients to score higher on “Positive affect” and lower on the “Feeling at home” subscale.

Conclusions: KS presents distinct QoL disparities when compared to dementia. Those with KS tend to experience richer social connections and heightened positive emotions than their dementia-afflicted counterparts. While dementia patients exhibit more restless tendencies, KS patients generally feel a lesser sense of belonging in nursing homes. The findings underscore the necessity for tailored nursing homes and care programs to meet the unique requirements of both patient groups.

Individuals diagnosed with KS often exhibit stronger social connections and a higher frequency of positive emotions compared to those with dementia. On the other hand, dementia patients tend to demonstrate increased restless behaviors in contrast to their KS counterparts. The variances between the two patient groups might stem from their distinct cognitive impairments. Specifically, KS is characterized by deep-seated amnesia and often executive dysfunction, necessitating structured and well-organized daily routines. Consequently, we advocate for tailored nursing homes and care plans designed to address the unique requirements of KS patients [68]. 

## 8. Patient Care, Nursing, and Mental Capacity

### 8.1. Assessment of the Patient

Evaluations should rule out other potential reasons for the patient’s symptoms. This could require drug tests, checks for anemia, leukemia, and variations in blood sugar levels. It is essential to conduct blood tests, including y-glutamyl transpeptidase, serum B1, pyruvate level, and an assessment of thiamine status by measuring erythrocyte transketolase activity, as these can assist in the diagnosis.

Furthermore, EEG is needed only when trying to monitor patients with seizures, while CT or MRI scans can identify brain alterations [69].

### 8.2. Management of the Patient

Whenever patients exhibit the described signs and symptoms, a potential diagnosis of Wernicke’s/Korsakoff’s should be taken into consideration, regardless of their age. These indicators might not always be clear-cut. If other health issues can be dismissed, a history of alcohol consumption should be obtained, if not already documented [70].

If WE or KS is confirmed, the primary treatment involves a high-dose thiamine injection, ideally intravenous, with a maximum limit of 300 mg for intense deficiencies. This replenishes the body’s essential vitamin B1 levels, potentially averting brain injury or even death. After this intervention, the patient’s condition should be re-evaluated. While the risk of an allergic response to high thiamine doses is minimal, nursing staff should be vigilant and ready to address anaphylactic reactions [71].

Daily electrolyte levels should be assessed and tracked for a duration of 7–10 days. Monitoring both food and fluid consumption and excretion is crucial. Due to the difficulties in metabolizing and absorbing vital nutrients caused by extended alcohol use, a treatment plan should incorporate parenteral nutrition [72].

It is essential to avoid triggering the “refeeding” syndrome. This occurs when patients suddenly consume high-protein calories beyond their standard capacity, potentially leading to cardiac issues [50].

Administering glucose before thiamine is discouraged, given the potential imbalances in electrolytes and fluids and a possible surge in insulin levels due to the abrupt nutritional influx.

## 9. Neuroimaging Research and Future Directions

### Advanced Imaging Techniques

Previous studies employing magnetic resonance imaging (MRI) to investigate alcoholism have predominantly focused on cerebral structures, often sidelining the cerebellum. However, advancements in neuroimaging techniques, particularly the utilization of voxel-based morphometry (VBM) [62], have expanded the scope of investigation to include a comprehensive analysis of multiple brain regions, such as the diencephalon, midbrain, and cerebellum. The efficacy of VBM has been substantiated in the examination of various neurological and psychiatric conditions, including schizophrenia [73], aging, and Alzheimer’s disease [74].

In the realm of alcoholism research, VBM has been employed to elucidate structural brain differences between alcohol-dependent and non-dependent controls [75]. One study conducted a comprehensive analysis of gray and white matter in a cohort of 22 alcohol-dependent individuals, contrasting their neuroanatomy with age- and gender-matched controls. The findings revealed marked reductions in gray matter volume in regions such as the thalamus, posterior hippocampus, and frontal cortical areas among the alcohol-dependent subjects. Concurrently, the study observed white matter atrophy in the pons and cerebellum. A subsequent investigation compared gray matter volumes in 45 abstinent alcoholics and 50 controls, identifying significant reductions in the lateral prefrontal cortex, medial frontal cortex, and posterior cingulate gyrus in the alcoholic group [75].

Such morphological alterations in gray matter have also been documented in other substance-dependent populations, such as cocaine, cannabis [76], methamphetamine, and heroin users [77]. However, research on polysubstance abusers remains limited [78]. Noteworthy studies in this niche have highlighted reduced gray matter volume in the bilateral prefrontal lobes, as identified by Liu et al. [77], and in the medial orbital frontal cortex, as observed by Tanabe et al. [78].

Prior research conducted at the NIH Clinical Center focused on the forebrain volumes of alcohol-dependent patients, categorizing them into monosubstance and poly-substance abusers [79]. This research revealed minimal differences in overall gray and white matter volumes between the two subcategories of alcoholics. Nonetheless, the study had limitations in its narrow focus on forebrain structures, excluding potential changes in the diencephalon or midbrain—regions previously implicated in alcoholism effects [69].

A more recent investigation employed VBM to conduct a nuanced examination of regional brain volumes between alcohol-dependent subjects and non-dependent controls. The study postulated that alcohol-dependent individuals would manifest reduced gray matter volume in specific brain regions, such as the frontal lobes and cerebellar cortex. The investigation also aimed to identify whether volume differences existed within subregions between alcoholics who were polysubstance abusers and those who were exclusively alcohol-dependent. Given the absence of global gray matter volume differences in prior studies [79], the study endeavored to unearth regional volume discrepancies.

The use of VBM is particularly advantageous for differentiating Wernicke-Korsakoff Syndrome (WKS) from general alcohol-related cognitive disorders. VBM’s capacity for detailed regional analysis enables a more nuanced understanding of the specific structural brain changes attributed to WKS. This level of specificity provides crucial insights into the distinct pathophysiology of WKS, thereby facilitating more targeted therapeutic interventions compared to broader alcohol-induced cognitive impairments.

## 10. Conclusions

Nearly a century and a half has elapsed since Carl Wernicke first identified the symptoms indicative of persistent thiamine deficiency. Despite this lengthy period, the condition often remains overlooked, underdiagnosed, and inadequately treated. The challenge of accurately diagnosing the condition arises from its symptoms, which closely mimic those of other conditions like acute alcohol intoxication. This diagnostic ambiguity makes rapid treatment paramount, often taking precedence over an exhaustive diagnosis. Intravenous thiamine replenishment remains the gold standard for quick treatment, although healthcare providers have traditionally been hesitant to administer it. For instance, a retrospective analysis found that merely one-fifth of head injury patients at risk of thiamine deficiency were actually treated with thiamine, and of these, the majority were given low doses orally for a limited duration. 

This hesitancy is compounded by misconceptions surrounding the rarity of the condition and the notion that a diagnosis requires the presence of a classic symptom triad. Swift and decisive treatment with intravenous thiamine is crucial to prevent irreversible brain damage due to biochemical and metabolic imbalances. An estimated 80% of those experiencing WE, the acute and treatable phase, may evolve into Korsakoff’s syndrome, a chronic condition marked by persistent memory impairments and potential confabulation. In the most severe cases, the syndrome can be fatal, accounting for approximately 20% of such cases. 

Though the optimal thiamine administration regimen remains to be definitively established, identifying at-risk individuals and debunking common myths about Wernicke-Korsakoff syndrome can play a significant role in reducing both morbidity and mortality in this vulnerable population. Future research should focus on fine-tuning thiamine administration protocols to improve further outcomes.

## Figures and Tables

**Figure 1 jcm-12-06101-f001:**
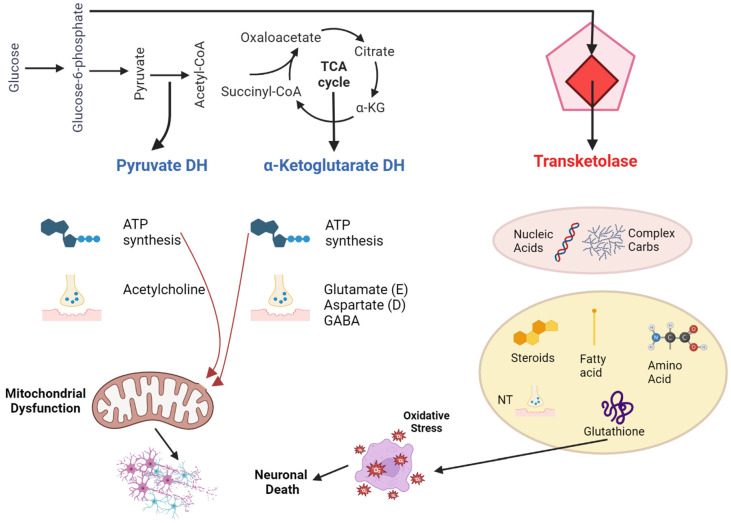
Pathophysiology of thiamine deficiency in Wernicke-Korsakoff syndrome.

**Table 1 jcm-12-06101-t001:** The operational criteria for the diagnosis of WE, as elaborated by Caine et al [8].

Symptom or Sign	As Evidenced by One or More of the Following
Nutritional deficiencies	– Undernutrition
	– A record of significantly impaired nutritional intake
	– An unusual thiamine status
Oculomotor impairments	– Nystagmus
	– Gaze palsy
	– Ophthalmoplegia
Cerebellar impairment	– Abnormalities of past pointing
	– Instability or ataxia
	– Dysdiadokokinesia
	– Impaired heel–shin testing
Either an altered mental state	– Disorientation in two of three fields
	– Confused
	– An abnormal digit span
	– Comatose
or	or
Mild memory dysfunction	– Failure to remember two or more words in the four-item memory test
	– Impairment on more elaborate neuropsychological tests of memory function

Notes: When two out of these four criteria apply, the clinical diagnosis of WE is made. Abbreviation: WE, Wernicke’s encephalopathy.

**Table 2 jcm-12-06101-t002:** Clinical Examples of Medical Comorbidities Associated with Thiamine Deficiency, Grouped by Mechanism.

Mechanism of Thiamine Deficiency	Etiology	Clinical Examples
Accelerated usage	Hypermetabolic state	Systemic illness Infection/sepsis Thyrotoxicosis Pregnancy
	Rapid cell turnover/high cell density	Hematological malignancy Fast-growing tumor
	Excess glucose metabolism	Seizures, following rapid influx of glucose Alcohol withdrawal
Impaired utilization	Decreased enzyme activity	Malignancy Co-factor deficiency Chemotherapy-induced
Increased losses	Iatrogenic	Hemodialysis
Decreased availability	Malabsorption	Bariatric surgery Crohn’s disease
	Malnutrition	Anorexia nervosa Hunger strike Fad dieting Bariatric surgery Homebound elderly TPN (if not supplemented)
	Starvation	Anorexia nervosa Hunger strike GI obstruction Chemotherapy Systemic illness
	Vomiting	Hyperemesis gravidarum Chemotherapy-induced Status post abdominal surgery GI obstruction Pancreatitis Bariatric surgery

**Table 3 jcm-12-06101-t003:** Comparison of Guidelines for Diagnosis and Treatment of Suspected WE.

	Who to Treat	Route	Frequency	Duration
European Federation of Neurological Societies (EFNS)	People with a combination of any two of the following: (1) inadequate nutrition (2) oculomotor abnormalities (3) cerebellar dysfunction (4) mild memory problem	IVpreferred	TID	Until the symptoms are resolved.
Royal College of Physicians (RCP)	Signs of alcohol abuse with one of the following: (1) acute confusion (2) reduced level of consciousness (3) ataxia (4) ophthalmoplegia (5) memory problems (6) hypothermia with hypotension	IV	TID	Three days. If a positive response is observed, continue with a daily dose of 250 mg administered intravenously or intramuscularly for 5 days, or until clinical improvement stops.

## Data Availability

Not applicable.

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
