# Peer review of "Neuropsychiatric and Neuropsychological Aspects of Alcohol-Related Cognitive Disorders: An In-Depth Review of Wernicke’s Encephalopathy and Korsakoff’s Syndrome"

_jcm, 2023, doi:10.3390/jcm12186101_

Round 1
Reviewer 1 Report
The text is interesting and considers two neuropsychological syndromes that may have a common pathogenetic mechanism related to alcoholism and vitamin B1 deficiency. The Authors after a historical picture describe the syndromes. Despite the interesting pathogenetic mechanism I believe the article should be edited before it is worthy of publication. The abstract could be shorter and provide conclusions. The presentation of cases I think may be unnecessary. The chapter of neuroimaging is too long; it could be limited to a table indicating the scientific works using this method in the two syndromes. The article as a whole comes across as too long. The conclusions are too brief.
Author Response
Dear Reviewers,
In response to the comprehensive feedback from the reviewers, we have undertaken extensive revisions to improve the clarity, coherence, and overall rigor of the review article. Below is a summary of the modifications made in accordance with each point raised:
Thank you truly for your time and expertise, alongside with your guidance.
We hope that we addressed your suggestions, the abstract is now shorter and better organized, we have deleted the presentation of cases, we have shorten the neuroimaging chapter and expanded on the conclusion. Moreover, we have made other improvements to the manuscript's quality.
Thank you for your meaningful comments!
- Figure 1 has been excised from the article as suggested.
- Consistency in the use of abbreviations for Wernicke's Encephalopathy (WE) and Korsakoff Syndrome (KS) has been maintained throughout the text.
- The repetitive content in the 'Criteria for Diagnosing Korsakoff' section has been reworked for conciseness and clarity.
- The sentence with the term "Kinesis" has been rectified to correctly state "Korsakoff."
- The last paragraph concerning the Caine criteria and Table 1 have been relocated under the heading 'Criteria for Diagnosing WE.'
- The paragraph discussing impairments commonly observed in WE patients has been moved to the 'Neuropsychiatric and Cognitive Sequelae' section.
- The two case studies have been removed, making the manuscript more easy to understand and read
- Repetitive and redundant paragraphs have been either revised or relocated. Notably, the content pertaining to short-term memory (STM) and confusional states has been moved to the 'Neuropsychiatric and Cognitive Sequelae' section.
- The 'Neuroimaging' section has undergone a complete overhaul for clarity, focusing on key MRI findings for WE and KS while omitting unnecessary details on MRI sequences.
- A new paragraph has been inserted in the 'Biomarkers' section to clarify the clinical versus research utility of markers such as CDT, EtG, FAEEs, and Peth.
- The 'Neurologic Manifestations of WE' section has been rebranded as 'Neurologic and Cognitive Sequelae' to more accurately reflect its content.
- Tables 2 and 3 have been reworked to eliminate redundancy and ensure that column headings accurately represent the data below them.
- The section on 'Effectiveness and Limitations of Nonpharmacological Interventions' has been revised for clarity, particularly in explaining the role of errorless procedural learning in episodic memory disorders.
- The heading for Section VIII has been renamed to 'Patient Care, Assessment, and Management,' to more accurately represent the scope of the content.
- The 'Assessment' section has been updated to clarify that an electroencephalogram (EEG) is not part of the standard evaluation and that hypernatremia testing falls under standard lab evaluation.
- The section discussing the utility of voxel-based morphometry (VBM) in research has been rewritten to provide a more nuanced explanation of how this imaging technique could specifically contribute to the understanding of Wernicke-Korsakoff Syndrome (WKS) as opposed to general alcohol-related cognitive disorders.
We hopr that these revisions substantially address the concerns and enhance the overall quality of the manuscript. Thank you for the constructive feedback, which has been invaluable in refining the article.
Reviewer 2 Report
Thank you for doing a comprehensive review of Wernicke Korsakoff syndrome. I think the review would benefit by a clearer synthesis of key points, better organization and less repetition to make it more clinically relevant.
In specific:
I would delete Figure 1 as I do not think it adds any information.
If you are going to use abbreviations such as WE or KS stick to them on don’t keep writing out Wernicke’s Encephalopathy and Korsakoff syndrome.
Under the Criteria for diagnosing Korsakoff paragraph 2 is repetitive and should be reworked
The sentence starting with “One objective … I think you mean Korsakoff rather than Kinesis
The last paragraph as regards the Caine criteria belongs under the heading criteria for diagnosing WE as does Table 1
The paragraph that starts with “WE patients frequently exhibit impairment …” belongs under the Neuropsychiatric and Cognitive Sequelae section.
I am not sure I see the point of the 2 cases. They don’t add to the review and are confusing as to how to interpret the neuropsychological testing. I would either leave them out or clearly delineate key points of their initial presentation and long-term outcomes.
The next paragraphs need to be rethought. The one starting with “WE typically manifests with…” is repetitive. You have made this point several times. The next several paragraphs discussing memory function what is you point? Is it that STM cannot be assessed in the setting of a confusional state? Shouldn’t most of this be under the Neuropsychiatric and Cognitive Sequelae section?
The neuroimaging section is confusing and needs to be rewritten. What are the key findings on MRI for WE and KS? You do not need to explain the physics of MRI sequences.
The biomarkers section includes many items that are never used in clinical practice such as CDT, EtG, FAEEs and Peth. They may be helpful research tools, but that should be clearly stated.
Under section III you again state the neurologic manifestations of WE. This is not necessary. The eye movement and cerebellar manifestations of WE are not neuropsychiatric manifestations of the disorder, so maybe the section should be renamed – Neurologic and Cognitive Sequelae
Table 2 needs to be reworked. There is a lot of repetition.
Table 3 also needs attention. It has many duplications and the column headings do not correlate with the information below.
The information under Effectiveness and limitations of nonpharmacological interventions needs to be clearer. You need to explain better how errorless procedural learning helps someone impaired by an episodic memory disorder.
Sectin VIII- I would relabel this Patient Care, assessment and management. Nurses don’t do this physicians do.
Under the assessment section- an EEG is only ordered when needed and is not part of the standard evaluation of these patients. The testing for hypernatraemia is covered under the standard lab evaluation.
Under the IX section- I appreciate the VBM may be a helpful research tool. However this section contains little as to how this technique might be helpful to understand WKS rather than alcohol related cognitive disorders.
no
Author Response
Dear Reviewers,
In response to the comprehensive feedback from the reviewers, we have undertaken extensive revisions to improve the clarity, coherence, and overall rigor of the review article. Below is a summary of the modifications made in accordance with each point raised:
Thank you truly for your time and expertise, alongside with your guidance.
We hope that we addressed your suggestions, the abstract is now shorter and better organized, we have deleted the presentation of cases, we have shorten the neuroimaging chapter and expanded on the conclusion. Moreover, we have made other improvements to the manuscript's quality.
Thank you for your meaningful comments!
- Figure 1 has been excised from the article as suggested.
- Consistency in the use of abbreviations for Wernicke's Encephalopathy (WE) and Korsakoff Syndrome (KS) has been maintained throughout the text.
- The repetitive content in the 'Criteria for Diagnosing Korsakoff' section has been reworked for conciseness and clarity.
- The sentence with the term "Kinesis" has been rectified to correctly state "Korsakoff."
- The last paragraph concerning the Caine criteria and Table 1 have been relocated under the heading 'Criteria for Diagnosing WE.'
- The paragraph discussing impairments commonly observed in WE patients has been moved to the 'Neuropsychiatric and Cognitive Sequelae' section.
- The two case studies have been removed, making the manuscript more easy to understand and read
- Repetitive and redundant paragraphs have been either revised or relocated. Notably, the content pertaining to short-term memory (STM) and confusional states has been moved to the 'Neuropsychiatric and Cognitive Sequelae' section.
- The 'Neuroimaging' section has undergone a complete overhaul for clarity, focusing on key MRI findings for WE and KS while omitting unnecessary details on MRI sequences.
- A new paragraph has been inserted in the 'Biomarkers' section to clarify the clinical versus research utility of markers such as CDT, EtG, FAEEs, and Peth.
- The 'Neurologic Manifestations of WE' section has been rebranded as 'Neurologic and Cognitive Sequelae' to more accurately reflect its content.
- Tables 2 and 3 have been reworked to eliminate redundancy and ensure that column headings accurately represent the data below them.
- The section on 'Effectiveness and Limitations of Nonpharmacological Interventions' has been revised for clarity, particularly in explaining the role of errorless procedural learning in episodic memory disorders.
- The heading for Section VIII has been renamed to 'Patient Care, Assessment, and Management,' to more accurately represent the scope of the content.
- The 'Assessment' section has been updated to clarify that an electroencephalogram (EEG) is not part of the standard evaluation and that hypernatremia testing falls under standard lab evaluation.
- The section discussing the utility of voxel-based morphometry (VBM) in research has been rewritten to provide a more nuanced explanation of how this imaging technique could specifically contribute to the understanding of Wernicke-Korsakoff Syndrome (WKS) as opposed to general alcohol-related cognitive disorders.
We hope that these revisions substantially address the concerns and enhance the overall quality of the manuscript. Thank you for the constructive feedback, which has been invaluable in refining the article.
Round 2
Reviewer 2 Report
Thank you for the much improved manuscript.
Author Response
Dear Reviewer,
Thank your for your time, expertise and important suggestions!
Our best regards!